# Integrated Psycho-Socio-Educational Programmes for Women Victims of Gender-Based Violence

**DOI:** 10.3390/healthcare12171795

**Published:** 2024-09-08

**Authors:** María Cristina Lopes-Dos-Santos, Sagrario Anaut-Bravo

**Affiliations:** Department of Sociology and Social Work, Public University of Navarra, 31006 Pamplona, Spain; sanaut@unavarra.es

**Keywords:** self-care, gender-based violence, integrated support, psycho-socio-educational intervention, empowerment

## Abstract

Although women victims of gender-based violence suffer health and self-care issues, there is still a lack of coordinated integrated support programmes aimed at mitigating the socio-health impact of gender-based violence on women that are suitably adapted to new social contexts. Given the importance of analysing the effectiveness of integrated psycho-socio-educational interventions and self-care programmes, this study reviews the interventions and programmes implemented in Spain, focusing specifically on their implementation and impact in the autonomous region of Navarra. The data reviewed were extracted from official publications and websites pertaining to the 17 Spanish autonomous regions and the corresponding Spanish government ministries. Additionally, 18 reports on psycho-socio-educational interventions (*n* = 63) for the whole of Spain centring on integrated support for self-care were analysed. In the case of the autonomous region of Navarra, four psycho-socio-educational intervention programmes (*n* = 4) involving self-care were analysed. The analysis found that there are a number of diverse integrated psycho-socio-educational programmes focusing on self-care that provide support and interventions. The results indicate that the women who benefit from the services, especially those relating to self-care (self-esteem, healthcare, personal image, and health), report improvements in their general wellbeing.

## 1. Introduction

There is a wealth of historical scientific literature focusing on the violence suffered by women in their domestic environments, primarily at the hands of their partners or ex-partners [1,2,3,4]. Similarly, there are also many studies on the interventions implemented to support women victims of gender-based violence (GBV), their results, and their effectiveness [5,6,7].

Although progress has been made in identifying different forms of abusive behaviour in order to delimit [8,9,10,11] and develop a variety of theoretical approaches [12,13,14,15,16,17] towards GBV, it has become clear that the issue is both multivariate and complex. This reality is framed within a renewed social [18,19] and eco-economic [20,21] perspective and interest [22,23,24,25], as well as the exponential prevalence of GBV [26].

Research in Spain highlights an increase in GBV and a myriad of regional intervention approaches [27]. The impact of GBV on women’s quality of life is often identified during healthcare appointments [28] and social settings, in which early detection is key to prevention [29,30]. A victim’s self-perception often leads to the normalisation of their experiences [30] and the consequent lack of actions aimed at changing their situations [31]. Some of the most important consequences of GBV on women are trauma and injuries, depression and anxiety, and substance use, with three times more women victims of GBV attending healthcare appointments than other women [26]. A total of 28.7% of women living in Spain aged 16 to 74 who have had a male partner have suffered some form of violence. More specifically, 50.3% have suffered physical injuries and 78.4% psychological distress [18]. As a consequence, legislative changes and social advances have given rise to the implementation of various resources and actions in Spain’s 17 autonomous regions (excluding the autonomous cities of Ceuta and Melilla) [32,33,34,35,36,37,38].

One of the results of targeting resources at preventing, mediating, and ending GBV is the implementation of efficient close-contact interventions to reduce its impact and provide tools to empower women [39,40,41,42,43,44]. A number of relatively recent studies have highlighted the effectiveness of group activities [45,46,47] as key factors in improving mental health, self-care, and personal growth.

In turn, studies in Spain have highlighted a series of practises for their effectiveness [48,49,50] in supporting women, including cognitive contextual therapies, contextual therapies, and various other programmes [51]. For example, cognitive behavioural therapies are particularly effective in reducing anxiety and stress. Contextual therapies focus on individual clients’ needs and are primarily effective in reducing mental anguish and the risk of suicide, as well as improving problematic behaviours, but have not led to improvements in quality of life [52]. However, at the time of writing, it seems that there are no comparable evidence-based studies on the impact of integrated psycho-socio-educational support programmes, which comprise the coordinated intervention of psychological, educational and social aspects, on the mental health and wellbeing of women victims of GBV. Neither are there any studies on the promotion of self-care aimed at coping and dealing with GBV.

Within this context, this study aims to address the following objectives: (1) identify the integrated psycho-socio-educational intervention programmes for women victims of GBV in the Spanish autonomous regions; (2) identify the programmes that have demonstrated effectiveness in mitigating the psycho-socio-educational factors perceived by women victims of GBV, and (3) identify the promotion of self-care in the programmes analysed. Hypothetically, the results of the analysis should provide evidence on integrated psycho-socio-educational interventions focusing on promoting self-care and mental health through programmes implemented by multidisciplinary teams.

## 2. Materials and Methods

In order to respond to the general objective of determining the effectiveness of comprehensive psycho-socio-educational support programmes aimed at women victims of GBV in Spain, a systematised review (SR) of the scientific literature on the subject was performed using the intervention programmes identified. In addition to focusing on analysing and exploring areas of knowledge and research fields, the SR was aimed at identifying current trends in the subject, as well as determining research gaps and opportunities. This provided a “state-of-the-art” review [53,54]. A search, evaluation, analysis, and synthesis of the programmes under study was performed using the SALSA framework [53].

Searches were performed in Scopus and Dialnet. However, searches in the former did not produce any results when applying the territorial criteria—Spain and the names of the 17 autonomous regions into which Spain is administratively subdivided (Figure 1). Google Scholar was also used in the search. The searches in Google Scholar and Dialnet were performed using a combination of terms and text words to establish the inclusion criteria: programmes, integrated, psychological, mental health, social, educational, support, multidisciplinary teams, women, victims, gender-based violence, and self-care.

To complete the process, official publications and websites pertaining to the competent bodies in healthcare and social services in each of the Spanish autonomous regions were consulted. The Spanish Ministry of Health and the Ministry of Equality websites were also consulted. The latter incorporate the Institute of Women and the Government Office against Gender-based Violence.

The SR included the revision of 45 documents (procedure manuals, reports, resources, and services) from the 17 autonomous regions into which Spain is administratively subdivided (Figure 1). In addition, 18 official reports on the integrated support available to women victims of GBV were also included. None of the documents reviewed came from the Ministry of Health or the Ministry of Equality, which highlights the fact that the management and implementation of intervention programmes aimed at GBV is the responsibility of the autonomous regions.

In view of the limited data obtained, emails were sent to the aforementioned organisations requesting specific data on their programmes. Only five autonomous regions responded: Castilla y León, Madrid, Aragón, Catalonia, and the region of Valencia. However, they replied that they had no additional data other than those which appeared on their respective websites and had no results on the effectiveness of their programmes. Navarra was the only region to respond with additional data, which are presented below.

In the first instance, the selection criteria used for the public documents analysed contemplated programmes that included the following: psychological intervention, social intervention and/or educational intervention, or a combination of them all (psycho-socio-educational intervention). In reference to Spain as a whole or its autonomous regions, the programmes under study all had to be in force in 2024. The review did not assess or consider the methodological quality of the programmes analysed.

The content of the psycho-socio-educational GBV intervention programmes was then analysed to determine their impact, and, lastly, a quantitative and qualitative synthesis of the results was obtained to determine whether they contained evidence of the impact of the programmes implemented.

The SR revealed that all of the psycho-socio-educational GBV intervention programmes are aimed exclusively at women, as established in Spanish law (Organic Law 1/2004, of 28 December, on Comprehensive Protection Measures against Gender-based Violence). In turn, the programmes are designed and implemented by the autonomous regions themselves, either directly or through private entities. Although they follow their own criteria, there are similarities between them that enable a comparative analysis of their strategic lines. It was only possible to determine the effectiveness of the intervention programmes in 4 of the 63 cases, all corresponding to the autonomous region of Navarra.

## 3. Results

The autonomous regions have included their policies on equality and support for women victims of GBV in their healthcare systems as part of their social services, given that both areas fall under their responsibility. As a result, they consider integrated psycho-socio-educational support as an intervention methodology. However, owing to the fact that access to the official websites of the competent bodies was restricted (*n* = 15) and/or because the programmes were implemented by private entities (*n* = 1) whose references could not be obtained, it was not possible to confirm the existence and development of multidisciplinary intervention programmes in detail. Consequently, it was largely impossible to determine the composition of the programmes or the results of their impacts. In effect, detailed information was only available in the case of the autonomous region of Navarra (*n* = 1) after requesting a more comprehensive breakdown from the regional government of Navarra itself. Subsequently, data were extracted on the professionals who comprise the teams, the procedures implemented, and the evaluation of the results.

### 3.1. Programmes and Resources Targeted at Supporting Women Victims of GBV in the Spanish Autonomous Regions

Every Autonomous Region has a public body responsible for supporting women victims of GBV (*n* = 17) as part of social services, which usually fall under the healthcare system. The wide variety of programmes and resources available is due to the autonomy of each individual region, as detailed below.

In Andalusia, the Andalusian Institute of Women has an integrated support and refuge service for women victims of GBV and their dependents, comprising emergency centres, refuges, and supervised housing. It provides specific programmes for psychological support and legal advice as well as support services, in addition to accompaniment and social support. The refuges provide information, counselling, and integrated support through a multidisciplinary professional team. This takes the form of 180 municipal Women’s Information Centres and 8 provincial Women’s Centres [55,56,57].

In Aragón, the Institute of Women (IAM) provides information, counselling, and psychological, educational, social, and legal services to women victims of GBV. There is a psychological support service, a family education service, and psychological as well as social counselling services with offices in the three provincial capitals, Huesca, Zaragoza, and Teruel. Zaragoza City Council also provides a social support service for information and advice on GBV issues to women living in the locality. The IAM also provides a family educator service that has a coordinator, three educators (one in each province), and a reinforcement educator. In addition, legal advice is provided by the corresponding Bar Association at the IAM headquarters and regional centres, as well as other dedicated services [58,59].

Integrated support for victims of GBV in Cantabria provides information and emergency telephone helplines, a legal advice for GBV telephone helpline (number 016), a Support and Protection of Victims of Gender-Based Violence (ATENPRO) telephone helpline, the Integrated Forensic Assessment Unit for Gender-Based Violence at the Cantabria Institute of Forensic Medicine (UVIVG), and information and support centres in town councils and municipal associations [60,61,62].

There are support centres for women victims of gender-based violence in the Canary Islands on each of the islands. They provide emergency services and immediate assistance, housing, and specialised intervention (social, legal, employment and educational) [63,64].

In the autonomous region of Castilla-La Mancha, the services and programmes specialising in GBV at the Institute of Women consist of telephone helplines, refuge resources, tele-translation services, and Women’s Centres. Individual interventions involve counselling, advice, and mediation services in all of the region’s municipalities. Moreover, the region advocates a community intervention programme with the whole population in order to raise awareness about equal opportunities. Interventions includes services in the areas of rights, psychology, employment, and social orientation [65,66,67,68,69].

The integrated support programme for victims of GBV in the autonomous region of Castilla y León comprises the figure of a case coordinator (an expert from the public administration) who is part of the Primary Care Social Action Team. Their role is to detect, manage, coordinate, and design a plan together with experts in GBV from second-level multidisciplinary teams from the local area (detection, emergency attention, assessment and integrated support plan, provision of benefits, and follow-up). They provide expert legal and psychological support through a helpline for women victims of GBV (number 012), free in-person legal advice, and an accompaniment service staffed by volunteers who have had similar experiences [70].

The autonomous region of Catalonia has introduced a nuance in the terminology of its services and programmes. Instead of using the term “gender-based violence”, the Professional Intervention Service (SIE) focuses on women victims of “sexist violence” to highlight the origin of the violence that impacts women (i.e., male roles). The SIE provides information, integrated support, and resources to women, as well as their children, who have suffered or are suffering from violence, or are in the process of recovery and adjustment. The service is staffed by multidisciplinary teams comprising experts in psychology, law, social work, social education, and cultural mediation [71,72,73,74,75].

As in the case of Catalonia, in the Balearic Islands there is a social support and accompaniment service for victims of male/sexist violence. This is a dedicated service and provides the following avenues of support: social telephone helpline for cases of male violence, information, advice, specific counselling and professional support, referrals to appropriate dedicated resources, face-to-face accompaniment, and emotional support [76,77,78,79].

Although the autonomous region of Madrid is ultimately responsible for its intervention programmes, it does not manage them directly but outsources them to private entities via specific contracts (agreements and temporary subsidies). The composition of the teams varies depending on the entities’ resources. There are residential support centres for women victims of GBV, non-residential psychosocial support centres (MIRA Programme), and several dedicated support centres. In addition, a Regional Observatory on Gender-based Violence is located in 55 municipal sites, and there is also a legal advice service [80,81].

The Institute of Women in the autonomous region of Valencia provides resources in 24 h centres, dedicated residential centres, and crisis centres. It has a programme (Alba) with nine professionals in social work, social education, psychology, and law that, on an itinerant basis, provides emergency accommodation to women in situations of sexual exploitation [82,83,84].

The Institute of Women in Extremadura provides a support network for women victims of GBV. It has Women’s Centres in its two provincial capitals, Cáceres and Badajoz, staffed by interdisciplinary teams who provide integrated recovery programmes for women victims of GBV and emergency psychological support (Pilar Project), as well as emergency legal support [85,86].

The autonomous region of Galicia has Women’s Information Centres in many municipalities, as well as online legal advice and women’s helplines. It has also implemented various programmes for improving equal opportunities and ending GBV, in addition to a psychological support programme [87,88,89].

The support and protection measures for victims of GBV in the autonomous region of La Rioja focus on information and counselling services, economic benefits, and legal, psychological, and social support. The region has a Women’s Advice Centre, an emergency social support programme, and an accommodation network, among other services [90].

The autonomous region of Murcia has implemented emergency support resources in centres staffed by interdisciplinary teams. It has developed a regional network of 29 dedicated support centres for victims of GBV, 4 refuges, and supervised housing [91,92,93].

The autonomous region of the Basque Country provides a wide range of services and programmes adapted to its complex administrative reality. Thus, the resources and services targeted at supporting women victims of GBV are structured on three levels: the Basque Regional Government, 3 Provincial Capital City Councils (Victoria-Gasteiz, Bilbao and Donostia-San Sebastián), and 251 Town Councils. The Basque Department of Equality, Justice, and Social Policies provides a victim support service staffed by professionals in law, social work, and psychology; a co-ordination centre for violence against women with three unspecified professionals; integrated accompaniment in legal proceedings with five unspecified professionals; an information and telephone helpline staffed by eight unspecified female professionals; and economic benefits advice staffed by three unspecified female professionals.

The Provincial Councils provide emergency reception services (62 unspecified female professionals), residential centres (51 unspecified female professionals), legal advice services (12 professionals), psychological support services (141 professionals), and specialised programmes. In the province of Álava, the services provided are psycho-socio-educational intervention and social accompaniment; in Bizkaia, a specialised family intervention programme; and in Guipuzcoa, residential centres for women at serious risk from a lack of protection and social exclusion, as well as victims of male violence, and a regional social emergency service. In total, 53 professionals, without always specifying their area of expertise.

Lastly, municipalities provide an information, assessment, and specialised diagnosis service with 239 professionals (to be determined). There are accommodation services (sheltered housing) in which 59 unspecified professionals work, as well as a legal counselling and psychological support service [94,95,96].

The Institute of Women in Asturias has 18 Women’s Advisory Centres providing legal advice. There is also a Centre for Integrated Attention for Women Victims of Gender-based Violence (Casa Malva), which provides multidisciplinary interventions in addition to a psychosocial support service [97,98].

Lastly, in the autonomous region of Navarra there is a wide range of services focusing on GBV in both residential and non-residential situations. In the area of psycho-socio-educational support, the Integrated Support for Victims of Gender-based Violence Teams (EAIV) are multidisciplinary and work with women in residential and non-residential environments [99,100,101,102].

### 3.2. Integrated Psycho-Socio-Educational Intervention Programmes for Women Victims of GBV in the Autonomous Region of Navarra

In the autonomous region of Navarra, the integrated psycho-social-educational intervention programmes developed by multidisciplinary teams are implemented by the EAIV in residential centres (emergency, refuge, sexual violence, and housing) and/or in women’s places of residence. The EAIV perform ongoing work in relation to women victims of GBV [103,104,105,106].

The multidisciplinary teams are staffed by three professional figures (social educator, clinical psychologist, and social worker) who work together in geographical areas structured by zones. As a result, their actions converge in individualised support programmes in which all three intervene depending on the moment and needs of a client. Thus, the teams implement integrated intervention models in which the different disciplines provide individualised intervention programmes.

The programmes studied (*n* = 4) follow a standardised/consensual process designed in 2021 (procedure manual), which is summarised in Figure 2. In some cases, the process is direct because women victims of GBV get in touch with EAIV, proactively seeking help, as they have prior knowledge about the programme (verbal, posters, media, or professionals). In other cases, the process is indirect or referred from other services such as primary healthcare, local social services, local police, etc. The intervention process begins from the point of contact with the three professionals indicated.

In addition to their role as team coordinators, social workers also implement social interventions. Together with the other team members, they participate in the reception of victims and assess the level of GBV suffered by each individual (structured interviews) in order to subsequently determine the psychological and/or educational as well as social support required. Moreover, social workers deal with contextual aspects, assess the situation of abuse, the gravity, and risks, diagnose deficiencies and social difficulties, inform about existing resources, provide counselling, determine personal, family, and contextual competences, and promote personal education in self-care.

Psychologists (clinical psychology) assess a victim’s mental health and deal with the experience of GBV via therapy. They focus on the traumatic and psycho-emotional aspects, accompanying and supporting the victim throughout the process to improve their wellbeing and self-care.

Social educators accompany victims in their training and development through socio-educational skill programmes, which promote self-care and personal development towards greater autonomy and control over their own lives. The aim is to promote stable and sustainable changes in the process of overcoming their situation. They also accompany clients to community resources and legal proceedings.

The team works together to decide the intervention needs of each individual victim during meetings. An intervention plan is designed and developed together with the victim (personalised support), so that the objectives and the subsequent process are agreed directly and are implemented as well as assessed during the support process. This gives rise to an individualised report.

The intervention plan is assessed on the basis of the agreed objectives and their scope throughout the process, taking into account the support provided by the different professionals and specifying the actions implemented by each discipline. The intervention is implemented using scientific methodologies, intervention models, and measures adapted to each case, such as AF-5 [107], BDI-II [108], EGEP-5 [109], and STAI [110]. Various therapeutic techniques are used, including life history [111], lifeline [112], projective techniques [113], storytelling [114], and personal diaries [115], which are designed in relation to each individual process and need [116].

The intervention process is governed and assessed using follow-up reports every six months, which are designed according to the agreed objectives in line with the circumstances of each individual victim, focusing on aspects of wellbeing, mental health, and self-care. The intervention process ends after a maximum period of 18 months, with a final report that highlights the main aspects, the goals achieved, any unresolved issues, and those that require new interventions with other professionals.

According to the sources consulted [90,100,101,102], 80% of the women who benefitted from the services provided—therapy, psycho-educational and educational groups focusing on exclusive thematic and/or self-care approaches managed by dedicated experts—stated that they had experienced subjective improvement in their emotional state and daily lives (evaluation survey). Moreover, in the year 2023, the data show that 46.94% of the 556 women who entered the programme ended the intervention process because the objectives were met [117]. Taking into account the fact that 5.58% ended the process due to referrals to other resources and 4.32% because they moved to another autonomous region, the percentage of women (who remained in Navarra and were not referred to other resources) who ended the programme because the objectives were met is even higher: 52.09% of 501. The women who remain in the service either leave voluntarily and/or need to return to it at another moment in time due to their internalisation process or unexpected circumstances.

## 4. Discussion

The 17 Spanish autonomous regions provide diverse resources and services aimed at supporting women victims of GBV. However, centralised coordination in specialised institutes or bodies is only provided in five autonomous regions: Andalusia, Aragón, Asturias, Valencia, and Extremadura. The development of preventive and basic support programmes in towns and districts is more limited and is only contemplated in the autonomous regions of Castilla-La Mancha, Castilla y León, and Galicia. In contrast, there are specialised services and centres in every one of the autonomous regions. Moreover, although the autonomous regions may have regional governments with differentiated political views, and dispersed or concentrated populations, there are no differences in the administrative organisation of services.

Most are residential resources (emergency centres, residential centres, and housing) and/or information and support resources (legal and psychological support). Current Spanish legislation [118] envisages the creation of sexual violence centres in each of the 50 Spanish provinces by 2024, which implies the operation of at least 50 centres. Notwithstanding, Navarra already has such a centre in operation [119].

In summary, the Spanish autonomous regions provide integrated support and multidisciplinary teams for women victims of GBV, although the services specified are generally provided separately, i.e., legal resources on the one hand and psychological resources on the other. Telephone helplines, emergency support, and legal as well as psychological services are provided in every region, but how they are implemented in society varies in intensity (for example, the Basque Country has more dedicated and greater resources) and extension (autonomous regions with a greater number of provinces such as Andalusia, Castilla-La Mancha, and Castilla y León).

Although the data lead to the conclusion that the importance of intervention processes focusing on women victims of GBV is acknowledged and that the autonomous regions undertake and promote their implementation, their diversity and the fact that there is currently no publicly available description of the processes (there are no official publications) imply that it cannot be determined whether or not integrated intervention programmes are actually based on unified psycho-socio-educational programmes. Consequently, neither is it possible to determine the effectiveness of their results. The methodological limitations highlight the need for further research and the publication of results in order to determine the suitability of the processes currently in place.

Moreover, the outsourcing of services and resources from the public to the private sector entails agreements with entities/companies that have their own processes and methodologies that lead to diverse forms of intervention. Furthermore, their internal processes and assessments are subject to the “privacy” clauses of each entity; therefore, how the results are reported depends on the requirements of the corresponding public administration, who might or might not request detailed evidence from the private companies to support the results based on their own internal parameters of efficiency and efficacy. These issues have given rise to limitations in this study due to the lack of access to projects, methodologies, and evaluations, which legally belong to the private managing bodies.

However, there is evidence that several international programmes for women victims of GBV have shown favourable results in psychological and therapeutic interventions, indicating that women who remain in the programme tend to follow the advice and recommendations provided and are satisfied with the results [120]. Therapeutic approaches give rise to improvements in emotional wellbeing, especially cognitive behavioural therapies that reduce victims’ symptomatology of GBV [121] and increase levels of social support by implementing intervention programmes in stages [122]. Integrated, professional counselling improves victims’ wellbeing, mental health, and coping strategies [123]. Personal empowerment programmes and relaxation practises reduce stress and depressive symptoms [124], and the relationship between an expert and a victim reduces symptoms of depression and PTSD through the mechanism of empowerment in a safe space [125].

Similar evidence in other contexts has also been detected in the case of Navarra, which leads to the conclusion that intervention programmes targeted at supporting women victims of GBV in a psychological–therapeutic setting, together with social support, counselling, and empowerment through professional interaction, reduce mental health problems and promote greater security, empowerment, wellbeing and self-care. It would be beneficial to have more data on the programmes implemented and their results from all the Spanish autonomous regions in order to investigate the practical applications of different multidisciplinary intervention programmes and determine their impact on women victims of GBV, which might lead to more large-scale, efficient implementation.

## 5. Conclusions

The only intervention programmes available for analysis in this study were provided by the autonomous region of Navarra. The results seem to indicate that psycho-socio-educational programmes provide efficient and competent integrated support to women victims of GBV, which implies that integrated multidisciplinary methodologies are effective. However, the lack of data on the procedures and results from other autonomous regions made it impossible to compare data and processes. As such, the results of this study cannot be generalised; therefore, other processes and methodologies, as well as their results, need to be analysed in order to design suitable intervention models.

The results obtained bring into question whether it is really possible to refer to public policies aimed at supporting women victims of GBV in line with the directive established by current Spanish legislation, given that there is limited access to the data generated by the autonomous regions and the state. Assuming that data are either non-existent or very limited, it would seem pertinent to instigate a systematised collection of the processes, methodologies, and results obtained in the programmes implemented across the board. In short, it seems evident that there is ample room for improvement in the Spanish autonomous regions’ social services in the area of support for women victims of GBV.

## Figures and Tables

**Figure 1 healthcare-12-01795-f001:**
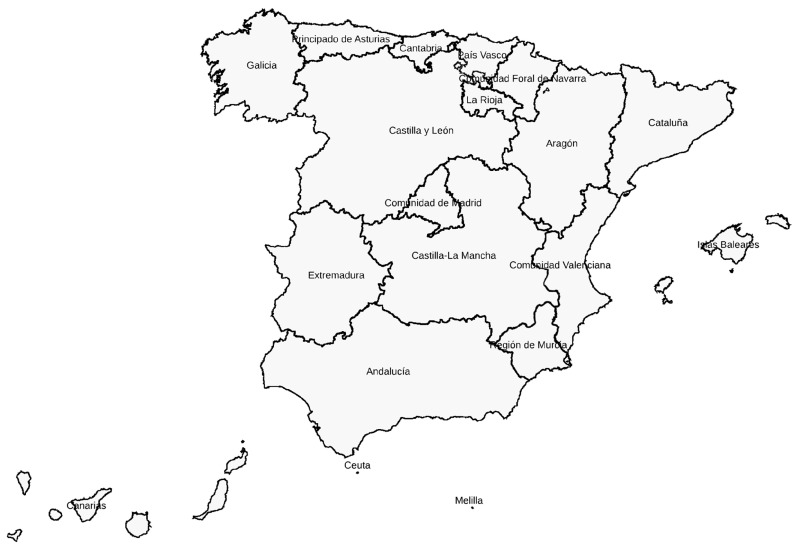
Map of the Spanish autonomous regions. Source: Google Maps.

**Figure 2 healthcare-12-01795-f002:**
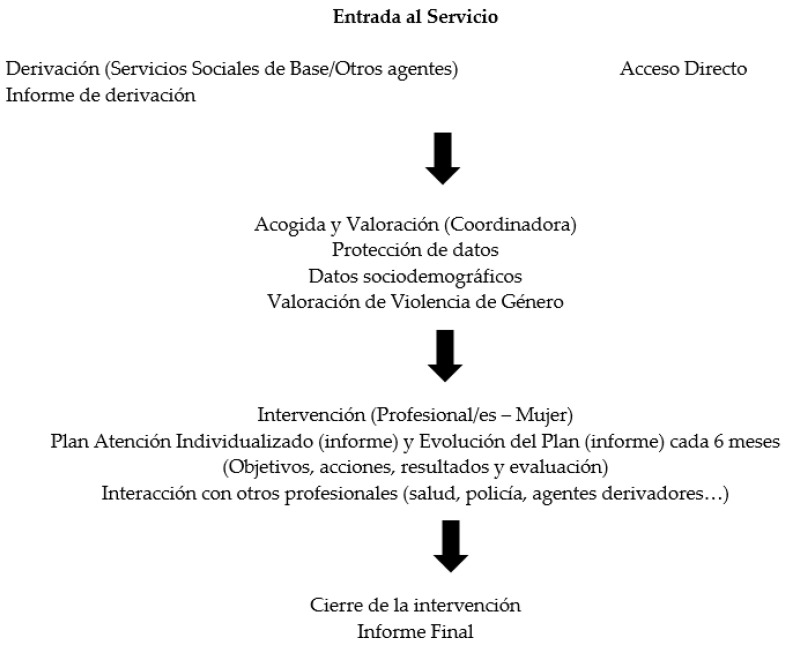
Support procedure for women victims of GBV by the EAIV in Navarra. Source: the authors.

## Data Availability

Data are contained within the article.

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
