# Peer review of "Integrated Psycho-Socio-Educational Programmes for Women Victims of Gender-Based Violence"

_healthcare, 2024, doi:10.3390/healthcare12171795_

Round 1
Reviewer 1 Report
Comments and Suggestions for Authors
There is no need for the last part of the title that says Self-Care Empowerment, it is addressed already by the Educational Programmes, and it can be deleted.
Comments on the Quality of English LanguageThe English were fine, however, it would have been nice to receive the article in Microsoft Word to make comments and suggestions over the document through track changes.
Author Response
Comment 1: There is no need for the last part of the title that says Self-Care Empowerment, it is addressed already by the Educational Programmes, and it can be deleted.
Answer 1: Thank you, very much! The last part of the title has been deleted.
The Materials and Methods section has been clarified and expanded.
The conclusions have been revised.
Comment 2: The English were fine, however, it would have been nice to receive the article in Microsoft Word to make comments and suggestions over the document through track changes.
Answer 2: Sorry, We have sent the revision in word.
Reviewer 2 Report
Comments and Suggestions for Authors
This is a well-written and well-structured paper. It would be useful to practitioners and commissioners of GBV services. It did not read particularly as an academic paper. The description of the Systematic Review process, for example, was thin. The presentation of findings was more a list than a synthesis. The method section mentions extraction of quantitative data, but this is descriptive only. There is no analysis of the effectiveness of interventions, but I think that is in part because the authors are saying such data is not available. Overall, it is a competent description of the issue, which may be useful to those working in the GBV sector.
Author Response
Comment 1: This is a well-written and well-structured paper. It would be useful to practitioners and commissioners of GBV services. It did not read particularly as an academic paper. The description of the Systematic Review process, for example, was thin. The presentation of findings was more a list than a synthesis. The method section mentions extraction of quantitative data, but this is descriptive only. There is no analysis of the effectiveness of interventions, but I think that is in part because the authors are saying such data is not available. Overall, it is a competent description of the issue, which may be useful to those working in the GBV sector.
Answer 1: Thank you, very much!
The Materials and Methods section has been clarified and expanded. Improvements include a more detailed description of the systematic review process and an improvement in the nature of quantitative data.
The conclusions have been revised in order to confirm and reinforce the reviewer's latest comments.
Reviewer 3 Report
Comments and Suggestions for Authors
This is an excellent article. It is scientifically sound, well conceptualised, clearly evidenced and its conclusions are framed by a prudent and reasonable understanding of the methodological (and substantive) limits to what can be said about the specific topic. The case study of Navarra indicates solid grounds for optimism about psycho-social-educational approaches and the authors carefully frame their recommendations for further research into this important question. The result is a discussion which is intrinsically interesting and socially important, both within the therapeutic and intervention field, and beyond.
Tiny point: lines 238 and 246 both introduce their paragraphs as "lastly". Perhaps use only once? ;)
Author Response
Comment 1:
This is an excellent article. It is scientifically sound, well conceptualised, clearly evidenced and its conclusions are framed by a prudent and reasonable understanding of the methodological (and substantive) limits to what can be said about the specific topic. The case study of Navarra indicates solid grounds for optimism about psycho-social-educational approaches and the authors carefully frame their recommendations for further research into this important question. The result is a discussion which is intrinsically interesting and socially important, both within the therapeutic and intervention field, and beyond.
Comment 2: Tiny point: lines 238 and 246 both introduce their paragraphs as "lastly". Perhaps use only once? ;)
Answer 1: Thank you, very much!
Answer 2:
It has been reviewed.
Reviewer 4 Report
Comments and Suggestions for Authors
Although the kickstarter of the paper was interesting, the presented methods are non-scientific, closer to a report than a scientific paper. The limitations are many:
Systematic Review is a rigorous and standardized approach that has not been executed, but instead a wide google research. It was not presented a research question, neither if PICO system was used. The Protocol was not presented with the criteria for selecting studies and methods for data collection (was PROSPETO used?).
The focus of the data collection was the implementation and impact of the programs in the Autonomous Region of Navarra, which are 4 in 63 programs analyzed.
The “effectiveness” of the programs was, in reality, the gathering conclusions of the results of the same programs, some of them lacking information (only the ones from Navarra Region were shared with the team, as it was pointed out as a limitation for the study). However, the effectiveness of a program enforces to follow a strict design in order to scientifically prove its benefits to the target population. Consequently the results are scarce and superficial.
Comments on the Quality of English LanguageMinor editing of English language required
Author Response
Comment 1: Systematic Review is a rigorous and standardized approach that has not been executed, but instead a wide google research. It was not presented a research question, neither if PICO system was used. The Protocol was not presented with the criteria for selecting studies and methods for data collection (was PROSPETO used?).
Answer 1: Thank you!
The Materials and Methods section has been clarified and expanded to answer and justify the systematic review, as well as to better specify the research question and the object of study. The criteria for inclusion/exclusion of data have also been included. The last paragraph contains the information required for the application of the PICO system.
The intention of the research was oriented, from the beginning, to analyse the different comprehensive intervention programmes in relation to gender violence in Spain in order to verify their effectiveness. However, the lack of elements of analysis that allow for conclusive rigour has led to results that are not very significant. Nevertheless, it is relevant to present these results as evidence of the need to establish evaluations on programmes of this type in order to orient practice towards scientific rigour and to advance in the effectiveness of intervention.
Comment 2: The focus of the data collection was the implementation and impact of the programs in the Autonomous Region of Navarra, which are 4 in 63 programs analyzed.
Answer 2: Data collection was the same for all the Autonomous Communities. The analysis of the 4 programmes allows us to know the results collected by these teams, as they do have data analysis and intervention results. It is not possible to make a comparison with other similar programmes, as there is no data about them. This limitation offers a clear vision of the need to establish reference values and results that show evidence and that allow for the improvement of programmes and teams for gender-based violence in Spain.
Comment 3: The “effectiveness” of the programs was, in reality, the gathering conclusions of the results of the same programs, some of them lacking information (only the ones from Navarra Region were shared with the team, as it was pointed out as a limitation for the study). However, the effectiveness of a program enforces to follow a strict design in order to scientifically prove its benefits to the target population. Consequently the results are scarce and superficial.
Answer 3:
The aim of the research was to understand the development, evolution and impact of the effectiveness of these programmes. Indeed, assessing these questions requires a scientific design and the corresponding evaluation of results. Lacking these elements, it has not been possible to ascertain these facts, except for the impacts referred to in the teams of the Autonomous Community of Navarre, which also require further monitoring and development. Nevertheless, we consider it relevant to show this reality as an important field for improvement in the implementation of projects in practical intervention, as well as in the field of research, which will enable the establishment of scientific base parameters for the development of these and similar programmes as a means of generating scientific evidence.
Comment 4: Minor editing of English language required
Answer 4:
It has been done.
Reviewer 5 Report
Comments and Suggestions for Authors
First of all, thank you for the opportunity to review this paper.
Although I consider the topic to be very important and the idea of the paper itself to be quite far-reaching, I have several major comments on the submitted manuscript that I offer to the authors and especially to the editors for consideration.
1. The authors write about GBV but focus only on women. In the scientific community, their output may contribute to a generalization (incorrect) that GBV victims are exclusively women, which is not true. I understand that the authors want to focus only on women, but if they want to use GBV, they should consistently explain the term and make it clear that it can refer to both women and men, as well as other genders, and therefore define that they want to focus only on women in their paper.
2. The authors state that early detection is key to prevention, but prevention precedes any prevalence, so rather than prevention being key to early intervention, the authors should be much more specific in this section (lines 42 and 43).
3. The authors write about an increase in GBV (line 40) but do not state over what period or the minimum percentage increase. An argument could be made here; the claim is thus not sufficiently supported.
4. The authors used Google for the search, they did not use any of the specialist databases. This is rather unusual for SR and reduces the scientific significance of the publication.
5. In lines 90- 92, the authors state that they included those programmes where only one of the aspects studied was included. How did the authors further distinguish between the programmes? Did they use a particular scoring scale? There is no further mention of this in the text.
6. It would certainly be useful to pay more attention to the inclusion and exclusion criteria for the programmes concerned.
7. From the fact that in their search the authors did not find appropriate programmes within the ministries mentioned (lines 102 - 104) I would be very cautious in the absence of the claim that the measures in question are not part of other documents in relation to the relatively narrow targeting of the search, where I do not recommend claiming on the basis of this fact alone, without further elaboration, that intervention programmes targeting GBV are the responsibility of the autonomous regions alone.
8. If the authors write about the effectiveness results (lines 113-114) and impact of the programmes, it would be good to indicate where they intend to obtain these, as this is very unlikely in terms of the programme and its measures, and they also do not refer in the given to relevant studies focusing on the impacts in question using the actors themselves.
9. If only one paper was found with a focus on the Region of Navarra, this is indicative of the fact that there is probably not enough information available to conceive an article for a peer-reviewed journal.
10. Figure 1 is completely irrelevant and redundant, lacking a source.
11. The remainder of the Results section does not meet the stated objectives of the study. It is only a descriptive description, which is certainly interesting, but lacks appropriate data condensation. The simple description does not meet the requirements of a scientific approach and as such cannot be considered as a result of the SR. The authors here deviate completely not only from their objectives but also from the cursory description of their methodology. As such, the text could perhaps be interesting for publication in one of the proceedings of a suitably themed conference, but in its current form, applying the existing approaches, it cannot be considered scientific and scholarly. For this reason I do not recommend the text for publication.
Author Response
Comment 1: The authors write about GBV but focus only on women. In the scientific community, their output may contribute to a generalization (incorrect) that GBV victims are exclusively women, which is not true. I understand that the authors want to focus only on women, but if they want to use GBV, they should consistently explain the term and make it clear that it can refer to both women and men, as well as other genders, and therefore define that they want to focus only on women in their paper.
Answer 1: Thank you, very much!
Legislation in Spain and its territories (Autonomous Communities) only refers to women when dealing with the issue of gender-based violence. The relevant clarification has been made in the text for the avoidance of doubt.
Comment 2: The authors state that early detection is key to prevention, but prevention precedes any prevalence, so rather than prevention being key to early intervention, the authors should be much more specific in this section (lines 42 and 43).
Answer 2: It has been clarified in the text, hopefully, more precisely.
Comment 3: The authors write about an increase in GBV (line 40) but do not state over what period or the minimum percentage increase. An argument could be made here; the claim is thus not sufficiently supported.
Answer 3: A bibliographic reference has been included [27]. Data from: State Observatory on Violence against Women of the Ministry of Equality (2022). XV Informe Anual del Observatorio Estatal de Violencia sobre la Mujer 2021. Madrid: Ministry of Equality. Available at: https://violenciagenero.igualdad.gob.es/violenciaEnCifras/observatorio/informesAnuales/informes/XV_Informe2021_Capitulos.htm
Comment 4: The authors used Google for the search, they did not use any of the specialist databases. This is rather unusual for SR and reduces the scientific significance of the publication.
Answer 4: The Materials and Methods section has been clarified and expanded to correct issues such as this.
Comment 5: In lines 90- 92, the authors state that they included those programmes where only one of the aspects studied was included. How did the authors further distinguish between the programmes? Did they use a particular scoring scale? There is no further mention of this in the text.
Answer 5:
The Materials and Methods section has been clarified and expanded: inclusion and exclusion criteria for the programmes have been incorporated. It was not possible to use any specific scoring scale, given that the determination of inclusion was based on the existence of the programme(s) and the presence of professionals in the social, educational and psychological fields, which is marked in the Spanish legislation as agents of comprehensive intervention in gender-based violence. Therefore, the inclusion of the programmes was marked by the existence of the programmes with the existing normative criteria.
Comment 6: It would certainly be useful to pay more attention to the inclusion and exclusion criteria for the programmes concerned.
Answer 6: The terms of exclusion and inclusion in these programmes are marked by the regulations (to whom they are addressed) and the competence of the Autonomous Communities for their implementation and development. Regulations on gender-based violence are mentioned in the text.
Comment 7: From the fact that in their search the authors did not find appropriate programmes within the ministries mentioned (lines 102 - 104) I would be very cautious in the absence of the claim that the measures in question are not part of other documents in relation to the relatively narrow targeting of the search, where I do not recommend claiming on the basis of this fact alone, without further elaboration, that intervention programmes targeting GBV are the responsibility of the autonomous regions alone.
Answer 7: With the clarifications included in the new Materials and Methods section, we believe we have answered this question.
Comment 8: If the authors write about the effectiveness results (lines 113-114) and impact of the programmes, it would be good to indicate where they intend to obtain these, as this is very unlikely in terms of the programme and its measures, and they also do not refer in the given to relevant studies focusing on the impacts in question using the actors themselves.
Answer 8: With the clarifications included in the new Materials and Methods section, we believe we have answered this question.
Comment 9: If only one paper was found with a focus on the Region of Navarra, this is indicative of the fact that there is probably not enough information available to conceive an article for a peer-reviewed journal.
Answer 9: There is no publicly accessible scientific publication with the results from Navarra. The information has been accessed through the results in the teams' reports and procedural manuals which explain how the service is developed, functions, procedures, intervention models, instruments and techniques, data records and data analysis. With the clarifications included in the new Materials and Methods section, we believe we have answered this question.
Comment 10: Figure 1 is completely irrelevant and redundant, lacking a source.
Answer 10: Figure 1 is relevant because it facilitates the understanding of the administrative organisation of Spain in Autonomous Communities of different size and composition. It is also relevant in order to understand similar policies and programmes between neighbouring Autonomous Communities. In other words, interventions on gender-based violence are visualised on the territory.
Comment 11: The remainder of the Results section does not meet the stated objectives of the study. It is only a descriptive description, which is certainly interesting, but lacks appropriate data condensation. The simple description does not meet the requirements of a scientific approach and as such cannot be considered as a result of the SR. The authors here deviate completely not only from their objectives but also from the cursory description of their methodology. As such, the text could perhaps be interesting for publication in one of the proceedings of a suitably themed conference, but in its current form, applying the existing approaches, it cannot be considered scientific and scholarly. For this reason I do not recommend the text for publication.
Answer 11:
We consider that the rigorous and scientific analysis that has been carried out, in addition to the improvements introduced in the sections on Materials and Methods and Conclusions, provides sufficient evidence on: limitations of access to the information generated (if any) by the Autonomous Communities and the State; possible absence of systematic collection of results achieved; predominance of criteria of compliance with the requirements of the regulations, but without conviction that the programmes implemented in each Autonomous Community are capable of adequately preventing and attending to cases of gender-based violence. In short, there is ample room for improvement in the social services systems in this area.
Round 2
Reviewer 4 Report
Comments and Suggestions for Authors
The intention of the research was oriented, from the beginning, to analyse the different comprehensive intervention programmes in relation to gender violence in Spain in order to verify their effectiveness - It was not an systematic review (as it is clear on the title). Even though it has been made an effort to strengthen the methodology, still it doesn´t fulfill the rigorous procedures that sustain an systematic review process starting with the core material, neither verifies their effectiveness.
Frequently, the results of psycho-socio-educational support programs targeting GBV victims’ are published in reports from governmental or non-governmental organizations that intervene with these victims, and are rarely published within an academic circuit, which difficult the aim of the this paper, as confirmed in lines 78-88. The research team decision in broadening the research considering nonscientific publications is comprehensible, but there was justification for this, neither how was it executed –“ official publications and websites pertaining to the competent bodies in healthcare and social services in each of the Spanish Autonomous Regions were consulted”
The method presented is very fragile and not consistent with an systematic review. Nevertheless the importance in understanding the quality of the intervention targeting GBV victims in order to map its strengths and vulnerabilities, what is present in this manuscript is a data collection of integrated psycho-socio-educational programs for women victims of gender-based violence implemented in Spain, that does not fulfill criteria for academic publication. Still, it is strongly suggested the publication of a report with these results in the native language and broadly disseminated in financing entities and structures that intervene with GBV victims, converging to a programme that addresses all the previous flaws and muscles the strengths.
Comments on the Quality of English LanguageThe quality of English Language is moderate, totally aceptable for non native speakers.
Author Response
In light of the reviewer's detailed feedback, we have included further clarification in order to validate the methodology and results. We have highlighted the gaps and lack of transparency in accessing information, as well as the importance of making the results of comprehensive psycho-socio-educational care programmes for women victims of gender-based violence in Spain public. If the results cannot be accessed or identified, then positive steps towards making social and healthcare policies to address gender-based violence cannot be implemented. Literature has been cited to corroborate the approach and results.
The paper has been proofread by a native English-speaking editor.
